# Providing Sustainable Knowledge for the Young Graduates of Economic and Social Sciences. Case Study: Comparative Analysis of Required Global Competences in Two Romanian Universities

**Lavinia Cornelia Butum** [1,*]**, Luminița Nicolescu** [1]**, Sergiu Octavian Stan** [2] **and Andrei Găitănaru** [2]

[1]    Faculty of International Business and Economics, The Bucharest University of Economic Studies, 010374 Bucharest, Romania; luminicolescu@yahoo.com
[2]    Faculty of Management and Faculty of Communication and Public Relations, The National University of Political Studies and Public Administration, 010643 Bucharest, Romania; sergiu.stan@facultateademanagement.ro (S.O.S.); agaitanaru@comunicare.ro (A.G.)
*    Correspondence: butum_lavinia@yahoo.com; Tel.: +40-722-382-207

**Abstract:** One of the most important challenges for our society is to keep a properly balanced insight of the relevant triangle of: Quality, functionality, and productivity. Regarding this, the most important challenge for universities now is to understand which are the main global competencies needed in the market, and to put them into practice in the educational process. Our main purpose in this paper is to identify the perception of students of the international competences, the personal competences, theoretical competences, practical competences, and others necessary competences needed for the national and international labor markets. For this purpose, a quantitative study was conducted, based on a survey addressed to 310 bachelor students, coming from two Romanian universities specialized in two domains: In social and in economic fields. The results present common and different views of the two groups of students, regarding specific competencies developed during studies and perceived to be needed on the labor market.

**Keywords:** global competences; internationalization; labor market; curricula; higher education

## 1. Introduction

In a global world, economies and citizens must be updated with technology, global standards, global professions, and global competences. Among the global professions, engineering represents the most standardized profession using mathematics, programming language, and English language [1]. Consequently, the technical universities have some advantages in providing competences that are required by both national and international labor markets. In the field of socio-economic studies, the situation is different, as there is less standardization involved. It is of interest to find out, from students' point of view, what are the activities that universities specialized in social studies on the one hand, and in economic studies on the other hand, have to undertake to provide the global competences for their students, given less standardized rules and norms in applying the theory into practice in these two fields.

The Theoretical framework section will highlight the implications of the changing economic environment on global markets stakeholders' activities and the role of internationalization for each stakeholder. The focus will be on the necessity for graduates, as future employees, to obtain global competences for both the national and the international labor markets. Additionally, the paper will

briefly address the activities developed in universities (that are driven by the internationalization process) for providing global competences to their graduates.

The methodology section comprises three different subchapters in order to clarify the constructs of the present research: Methods and statistical techniques used in research, the research variables, and the hypotheses proposed to be tested. There are seven research questions that represent the starting point of the research and search for students' perceptions on: Skills and abilities needed for students to obtain global competences (RQ1); skills needed for them to activate in the national labor market (RQ2), or to activate in the international labor market (RQ3); activities specific to the universities which contribute to the development of the skills needed for the national labor market (RQ4), or contribute for skills needed for the international labor market (RQ5); skills and abilities perceived by students as being ensured by their university/faculty at a general level (RQ6), or about differences between opinions of students from the two different faculties with different specializations from the two universities (RQ7).

The Results section presents on one hand the mean calculation for each variable for the two groups of students, and on the other hand validates the hypotheses. To validate the hypotheses, the significant differences based on *t*-test between students' perceptions in the two universities/faculties are tested (Hypotheses 1, 2, and 3) and the correlations between skills and university's/faculty's activities (Hypothesis 4).

The Discussion and Conclusions section reveals aspects connected to global competences that relate to other studies' findings and that show, in general, that the two groups of students with two study specializations have both similar, but also different, opinions regarding the skills and abilities needed to gain global competences. They have a similar view related to personal competences, workplace competences, and career management competences, and different opinions related to international competences, theoretical competences, and practical competences. In terms of competences provided by their universities/faculties for the national and international labor markets, the opinions converged only for the theoretical competences. All the other competences are perceived differently by students depending more on their specialization.

## 2. Theoretical Framework

Nowadays, the technology is making possible the communication and the exchange of information at international level. In this new environment, the general purpose of internationalization is to ensure a better connection of institutions and to provide more relevant services to society under changing realities [2]. Since stakeholders of global markets are expanding their activities beyond their local context, it is important to review briefly their search options that will be transformed into actions for accomplishing their quality and competitiveness goals in an internationalized environment:

(1)　Companies (either small national companies, or international companies) are in search for their best resources for development, their best place on market, and new models of business [3–6];

(2)　employees are in search of their best job opportunities even beyond national boundaries [7];

(3)　universities are in search of new students, new research opportunities, and of increasing the quality of their educational offers by providing graduates' employability [8–10];

(4)　governments are in search developing new programs and strategies, being active players in ensuring the politics, and regulation for the optimal function of economies and the citizens' well-being [11,12].

Under the above circumstances, the internationalization process is approached differently from the perspective of each stakeholder. For companies, internationalization means, on the one hand, to expand their business beyond national boundaries and, on the other hand, to adapt their activity and to provide goods and services that are valuable both at national and international level.

For the citizens and the work force, internationalization means competition in a global marketplace and the need to achieve sustainable competences for the world of work [7]. The current economic

climate has heightened awareness of the necessity to provide students' employability at graduation for local and international markets [10,13], supporting the idea that "citizens need to be engaged in issues and actions beyond their local context" [7] (p. 257), [14].

For higher education, "internationalization is understood as a general term for a broad range of activities in teaching and research that take place by crossing borders as well as locally in cooperation with international partners and students" [15] (p. 232). Depending on their location, these internationalization activities are divided in two areas: Internationalization "at home" and internationalization "abroad" [16]. The term "internationalization at home" includes a multitude of activities such as: Reviewing the content of the course and program from an international perspective, using foreign languages in the process of teaching and learning, using specialists from local or international companies and lecturers from abroad, and organizing extracurricular activities for students, to provide relevant competences for the students [16,17]. The term "internationalization abroad" is related to mobility of students, teachers, administrative staff, and curriculum, usually based on international agreements between universities [16].

From governments' point of view, internationalization is understood as a global phenomenon that needs to be met with a legal framework adapted to the changing environment to ensure the improvement of labor force and local economic competitiveness in a global marketplace [2]. In education, the governments have outlined strategies for the internationalization of higher education (HE) to "encourage direct engagement between industry and university and to promote economic growth and development" [9] (p. 22). These strategies and their specific activities are more indicative than mandatory for HE institutions, their implementation being adapted in each university according to their mission and their goals [12,18].

Thus, internationalization is seen as a global phenomenon that implies activities and involvement of all global stakeholders. The sustainable development in the economy starts from HE capacity to act as an agent for socio-economic progress, preparing highly skilled graduates that will continue to drive economic development both in local and international companies [9]. Additionally, the diversification of economic climate has brought into discussion the need for graduates to obtain various professionally relevant competences that will last a lifetime or at least 10 years upon graduation [19,20], and to benefit from the services provided by universities in job orientation and job search process [8,21]. Literature recognizes that the relevant competences are those required by the labor market upon employment and used at work in daily activity [13,22].

The literature has developed a detailed analysis of relevant competences depending on their level of recognition in the labor market, and one such typology includes: Theoretical competences, social and practical competences [23]. The theoretical competence or professional competence is an aggregate of theoretical notions (knowledge of professional terminology, theories, and concepts related to a specific profession) and critical thinking, called cognitive abilities [24] (p. 392), [25,26]. The social competences are related to individuals' ability to contribute through enthusiasm, communication, respect towards team members, and personal motivation in fulfilling the specific tasks of the workplace. This competence also involves relational skills, motivational skills, perception skills, and linguistic skills [23,27–29]. Additionally, a recent report [30] highlighted that non-cognitive capabilities or general skills are needed across all professions. From this category of cross-functional skills, the most demanded skills are: Leadership, communication, negotiation, creativity, problem solving [31,32], collaborative work, the use of technology to deliver materials [33,34], and cultural intelligence needed to succeed in a cross-cultural environment [35] as presented by recent studies. The practical competence or the functional competence refers to individuals' ability to apply in practice theoretical concepts and professional knowledge [23]. The sum of all these competences is intended to enable individuals to adapt to the changing demands of international labor market [9]. Competences are seen in the literature from different perspectives: The students and/or graduates [22,36–38], the university [39,40], and the employers' perspectives [22,33,41], and all of them have been considered when defining competences in the present study.

By developing the relevant skills for the future jobs and providing employability competences, the HE institutions contribute to economic growth and sustainable development [31]. Under these circumstances, literature has developed a new term: The global competences that represent "an overarching frame of reference encompassing multiple cognitive and non-cognitive components" [42] (p. 4). The global competences represent a complex learning goal being composed of a range of measurable competences: Analytical and critical skills or personal skills, knowledge and understanding or professional skills, attitude skills or relational skills, and international competences [42], applied knowledge or practical skills and workplace skills [43]. In a similar view, [44] are defining the global competence as a combination of self-awareness (ability to differentiate and mitigate risk in global environmental or social problems, ability to prosper and develop in any culture or country), intercultural communication (ability to adapt to other people from different cultures or with a different cultural backgrounds, ability to work and mediate interaction between people from different cultures, ability to speak more than one language fluently) and global knowledge (knowledge about issues that impact international relationships, ability to express personal views regarding global problems). Thus, the global competences represent a sum of: Knowledge and understanding of global issues, effective communication in more than one language, the ability to communicate with people from other cultures, and the ability to evaluate and act in different global and cultural circumstances [42,45].

Synthesizing the definitions given by the literature, we can divide the global competences into six main groups of competences: International competences, personal competences, competences related to career management, workplace competences, theoretical competences, and practical competences. Both classical and recent studies on skills, abilities, competences, and employability were used for this purpose as presented in Table 1. The study of the literature on skills, abilities, and competences showed that their definitions and approaches are very diverse, sometimes controversial, but they also present many commonalities. The perspective used to build the present typology was based on two guidelines: The inclusion of a large range of skills and abilities that contribute to individual employability and an extant agreement of the literature on what each category of competencies comprises. The present paper aims to highlight the students' perception on global competences detailed on the six groups. Table 1 details the six global competences and the skills and abilities associated to each global category of competence, as found in the literature and as used in the present research.

**Table 1.** Global competences.

| Global Competences | Skills and Abilities | Bibliographical References |
|---|---|---|
| International competences | Ability to communicate and to work efficiently in teams<br>Thorough knowledge of English language<br>Thorough knowledge of a foreign language other than English<br>Understanding the international political, economic, and business context<br>Ability to work in multidisciplinary teams<br>Ability to work in multicultural teams | [33,35,42,44,46] |
| Personal competences | Integrity<br>Adaptability<br>Initiative and involvement | [10,23,27,28,32,37,43] |
| Competences related to career management | Learning about the labor market and employment opportunities<br>Use of the Internet and recruitment platforms to find a job<br>Ability to present and to produce a CV | [10,29,36,43,47,48] |
| Workplace competences | Planning<br>Organization<br>Problem solving<br>Orientation towards customers and results<br>Critical thinking in decision making process | [10,31,43,48,49] |

**Table 1.** *Cont.*

| Global Competences | Skills and Abilities | Bibliographical References |
|---|---|---|
| Theoretical competences | Knowledge of professional terminology, theories, and concepts related to your profession | [10,22,24–26,33] |
| Practical competences | Correct use of principles and theoretical knowledge in practice<br>Use of information and technology (ICT)<br>Use of mathematics to solve operational problems<br>Knowledge of general economic theoretical concepts and formation of an economic way of thinking<br>Understanding the business reality | [9,22,23,33,37,50] |

Sources: Authors' compilation.

## 3. Methodology

### 3.1. Research Method and Statistical Techniques

In the present research, a survey was conducted using a questionnaire designed to identify the students' perceptions on global competences and on universities'/faculties' activities for providing the competences needed on the national/international labor market. The survey was used as a data collection method, as it allows a large number of responses and possible generalizations of results via statistical analysis [51] (p. 91).

The sample was formed of two categories of students from two different study specializations, one that through its profile is oriented more towards internationalization, and another that it is not. Besides identifying the perceptions on skills and competences for each group on its own, a comparison between the two groups of students was also envisaged.

One approach to statistical analysis consisted in descriptive statistics, used to measure the central tendencies [52] (p. 163) of the two groups of students. Another approach to statistical analysis referred to inferential statistics that included correlations on the one hand, and *t*-test for group differences on the other hand, as a method that allows for group comparison [51,52]. The SPSS statistical package was used for data analysis.

### 3.2. Research Variables

This paper presents the results of a survey conducted online with a total number of 310 bachelor students from two Romanian universities during October and November 2019. The first group is formed of 170 students from the National University of Political Science and Public Administration—Faculty of Communication and Public Relations (NUPSPA—CPR), and the second group is formed of 140 students from Bucharest University of Economic Studies—Faculty of International Business and Economics (BUES—IBE). The two faculties are both operating in the socio-economic field, one being specialized in social studies, the other in economic studies.

The study aims to identify similarities and differences in perceptions of the students with the two different specialization. The two categories of variables analyzed are: (a) Global competences as perceived by students, and (b) activities developed in universities for providing global competences for students. See Table 2.

**Table 2.** The research variables.

| The Research Variable | Description of the Research Variable |
| --- | --- |
| Global competences | A combination of theoretical, practical, personal, relational, international, and career-related management skills and abilities needed for the national and international labor market. |
| Activities carried out in university as part of the internationalization of higher education (HE) (for the development of global competences) | All activities related to internationalization at home and internationalization abroad: Activities focused on the internationalization of the curriculum; extracurricular activities promoted by the university for interconnection with groups belonging to local and international communities; activities focused on cross-border education. |

Source: Authors' own research.

The first category of variables the global competences looks at skills and abilities from three perspectives:

(a)   What are the skills and abilities needed to build global competences as perceived by students;
(b)   what are the skills and abilities needed to activate on the national labor market, as perceived by students;
(c)   what are the skills and abilities needed to activate on the international labor market, as perceived by students.

The second category of variables comprises the activities developed by universities with the two selected faculties for providing global competences (as skills and abilities) needed for the local labor market in comparison to the international labor market.

The online survey was distributed via Google forms. Students were invited to participate in the survey on a voluntary basis, and they were provided the survey link https://forms.gle/fcyrR7tMq7iGMvMQ6 with the estimated duration for completing the questionnaire of approximate 10–15 min. The questionnaire included 10 questions: 3 for identification, and 5 closed ended questions designed for providing information for our research questions. For each item in the questionnaire, we used the Likert five-step measurement scale, where 1 = to a very small extent/very low utility, and 5 = to a very large extent/very high utility.

The research questions addressed in the present study are:

RQ1: Which are the skills and abilities needed to obtain global competences?

RQ2: To what extent does the university ensure the development of the skills needed to activate in the national labor market?

RQ3: To what extent does the university ensure the development of the skills needed to activate in the international labor market?

RQ4: What are the university's activities that contribute to the development of the skills for the national labor market?

RQ5: What are the university's activities that contribute to the development of the skills for the international labor market?

RQ6: How do the skills and abilities perceived by students as being ensured by their university/faculty at a general level correlate to the contributions that specific activities conducted in the university/faculty have to the development of skills and abilities?

RQ7: Are there any differences between opinions of students from the two different faculties with different specializations (from the two universities)?

The study envisages to make a comparison between the perceptions of students of different socio-economic specializations in relationship to global competences and the ways their universities contribute to the development of these global competences on overall and via specific university activities.

*3.3. Hypothesis Testing*

Based on the theoretical considerations highlighted above and starting from the research questions, four research hypotheses are proposed in the present study:

**Hypothesis 1.** *The students' perception on "skills and abilities needed to obtain global competencies" present statistically significant differences between the two universities/faculties with different specializations.*

**Hypothesis 2.** *The students' perception on "skills and abilities provided by the university for the national labor market" present statistically significant differences between the two universities/faculties with different specializations.*

**Hypothesis 3.** *The students' perception on "skills and abilities provided by the university for the international labor market" present statistically significant differences between the two universities/faculties with different specializations.*

**Hypothesis 4.** *There are moderate to strong correlations between universities' activities and the skills and abilities provided by universities for the national/international labor market.*

For the testing of the hypotheses, we included the skills and abilities detailed in Table 1, that belong to different categories of skills and abilities that correspond to the following global competences: International competences, personal competences, competences related to career management, workplace competences, theoretical competences, and practical competences.

*3.4. The Research Instrument*

The consistency of statistical data was analyzed, for the whole lot of 310 students, as well as for the data subsamples from the faculties (NUPSPA—CPR for 170 students, respectively BUES—IBE for 140 students).

Additionally, the validity and the reliability of the questionnaire was tested. The validity of the questionnaire that refers to the degree to which the proposed measurement measures what it wanted to measure [53] was ensured at the questionnaire conception phase through the checking of the draft questionnaire with a number of experts from higher education. Validity is also proved by demonstrating the veracity of the hypotheses [53,54]. The reliability of the questionnaire is verified by calculating the Alpha Cronbach for the two groups and for the entire population included in the study [55,56]. Calculated values for Alpha Cronbach reliability test are $\alpha = 0.985$ for whole data collectively, $\alpha = 0.984$ for NUPSPA—CPR collectively and $\alpha = 0.983$ for BUES—IBE collectively. It can be concluded based on the Alpha Cronbach values, that there is a high level of statistical reliability, both for the entire data set and for the statistical data (two data subsets) analyzed at the university/faculty level.

## 4. Results

This section focuses on the presentation of the main results of the study.

*4.1. Means*

Means for all items were calculated and they are presented in Tables 3 and 4. For the first category of variables (global competences) (see Table 3), the results illustrate that the students from NUPSPA—CPR give more importance to the following groups of competences: Personal, relational, and career management competences than to the group of competences including professional competences and international competences (except for English knowledge). As part of different competences categories, very high averages (above 4.50) were recorded for skills such as: Adaptability (4.71), ability to communicate and to work efficiently in teams, English language, initiative and involvement (4.58), ability to produce and present a CV (4.55), organization (4.54), and integrity (4.51). The least

necessary skills required for building global competences were considered to be mathematics (3.06) and ability to communicate in foreign languages other than English (3.34).

Students from BUES—IBE more appreciate the international competences group, also paying special importance to other categories of competences: Personal, relational, practical, professional, and career management competences. For this group of students, the most important skill to develop global competences is represented by very good knowledge of English (4.78), followed by adaptability (4.66). Other skills that recorded high averages (over 4.50) for this group of students were: Ability to work in teams (4.54), initiative and involvement (4.53), and organization (4.51).

The analysis of the perception of students of the extent to which their university contributes to the development of skills necessary on the national and international labor markets depicts different results from one group of students to the other (see Table 3). However, both groups of students consider that the skills provided by their university are more useful to activate on national than on international labor markets. Thus, students from NUPSPA—CPR, perceive that their university allows them to develop skills necessary in the national labor market to a high extent for the following skills: Ability to communicate and to work efficiently in teams (4.36), initiative and involvement (4.35), adaptability (4.28), organization and integrity (4.24), planning (4.15), and problem solving (4.10), while the students from BUES—IBE appreciate that their university provides, to a high extent, the following skills necessary on the national labor market: Thorough knowledge of English language (4.26), knowledge regarding general economic theoretical and formation of an economic way of thinking (4.16), knowledge of professional terminology, theories, and concepts related to the profession and understanding of the international political, economic, and business contexts (4.07).

In respect to the skills needed to activate in the labor market, in both universities/faculties most skills (except for the English and other foreign languages skills) were perceived by students to be less developed (by their universities/faculties) for the international market in comparison to their development for the national market.

Information in Table 3 and associated comments contribute to answering the research questions RQ1 (Which are the skills and abilities needed to obtain global competences?); RQ2 (To what extent does the university ensure the development of the skills needed to activate in the national labor market?) and RQ3 (To what extent does the university ensure the development of the skills needed to activate in the international labor market?).

Table 4 presents the perceptions of students on the different universities'/faculties' activities and their perceived contribution to the development of skills and competences required on both the national and the international labor markets.

It can be noticed that for students from BUES—IBE, the activities conducted by the university that contribute to the development of skills and competences required on the labor market (both national and international) are: The intensive English courses provided by curricula (4.08/4.18), the students' mobility (3.99/3.96), international topics included in the curricula (3.74/3.76, curricula that lead to internationally recognized professions (3.69/3.84). At the same time, NUPSPA—CPR students consider that the main university activities through which skills and global competences can be obtained for both national and international labor markets are: Students' mobility (3.94/3.91), attracting students, researchers, and foreign educational specialists to develop new programs and to provide new skills (3.89/3.76), and extracurricular activities (3.86/3.82). Overall, students from BUES—IBE program (as compared to students from NUPSPA—CPR) perceive that their university prepares them better in terms of skills and global competences required on the labor market, as they score higher than the other group on average. At the same time, students from the economic field (BUES—IBE program) perceive that the different university's/faculty's activities prepare them better for the international labor market than for the national labor market. In case of this group of students, this can also be explained by the profile of the studies, international economic relations. In the case of the students from the social specialization (NUPSPA—CPR program) the results are mixed.

**Table 3.** Students' perceptions on skills needed for the development of global competences. Skills developed through the higher education program and their utility for national versus international labor market (averages).

| Skills and Abilities Part of Global Competences | Skills and Abilities Needed to Obtain Global Competences | | The University Ensures the Development of the Skills Needed to Activate in the National Labor Market | | The University Ensures the Development of the Skills Needed to Activate in the International Labor Market | |
|---|---|---|---|---|---|---|
| | NUPSPA—CPR | BUES—IBE | NUPSPA—CPR | BUES—IBE | NUPSPA—CPR | BUES—IBE |
| **International competences:** | | | | | | |
| Ability to communicate and to work efficiently in teams | 4.58 | 4.54 | 4.36 | 3.96 | 4.25 | 3.94 |
| Thorough knowledge of English language | 4.58 | 4.78 | 3.77 | 4.26 | 3.89 | 4.24 |
| Thorough knowledge of a foreign language other than English | 3.34 | 4.17 | 2.82 | 3.56 | 2.98 | 3.56 |
| Understanding the international political, economic, and business context | 3.84 | 4.44 | 3.85 | 4.07 | 3.75 | 3.88 |
| Ability to work in multidisciplinary teams | 4.09 | 4.37 | 4.05 | 3.59 | 3.92 | 3.48 |
| Ability to work in multicultural teams | 4.19 | 4.33 | 3.58 | 3.41 | 3.69 | 3.37 |
| **Personal competences:** | | | | | | |
| Integrity | 4.51 | 4.40 | 4.24 | 3.81 | 4.12 | 3.71 |
| Adaptability | 4.71 | 4.66 | 4.28 | 3.89 | 4.16 | 3.79 |
| Initiative and involvement | 4.58 | 4.53 | 4.35 | 3.94 | 4.18 | 3.78 |
| **Competences related to career management:** | | | | | | |
| Learning about the labor market and employment opportunities | 3.93 | 4.23 | 3.89 | 3.53 | 3.89 | 3.22 |
| Use of the Internet and recruitment platforms to find a job | 4.24 | 4.34 | 3.72 | 3.43 | 3.72 | 3.21 |
| Ability to present and to produce a CV | 4.55 | 4.47 | 3.76 | 3.44 | 3.75 | 3.25 |
| **Workplace competences:** | | | | | | |
| Planning | 4.38 | 4.36 | 4.15 | 3.74 | 4.09 | 3.60 |
| Organization | 4.54 | 4.51 | 4.24 | 3.83 | 4.15 | 3.63 |
| Problem solving | 4.48 | 4.46 | 4.10 | 3.74 | 4.01 | 3.53 |
| Orientation towards customers and results | 4.32 | 4.33 | 4.05 | 3.53 | 3.92 | 3.41 |
| Critical thinking in decision making process | 4.35 | 4.39 | 4.12 | 3.65 | 4.02 | 3.59 |

**Table 3.** *Cont.*

| Skills and Abilities Part of Global Competences | Skills and Abilities Needed to Obtain Global Competences | | The University Ensures the Development of the Skills Needed to Activate in the National Labor Market | | The University Ensures the Development of the Skills Needed to Activate in the International Labor Market | |
|---|---|---|---|---|---|---|
| | NUPSPA—CPR | BUES—IBE | NUPSPA—CPR | BUES—IBE | NUPSPA—CPR | BUES—IBE |
| **Theoretical competences:** | | | | | | |
| Knowledge of general economic theoretical concepts and formation of an economic way of thinking | 3.83 | 4.29 | 3.95 | 4.16 | 3.88 | 3.97 |
| Knowledge of professional terminology, theories, and concepts related to profession | 3.86 | 4.18 | 4.03 | 4.07 | 3.89 | 3.94 |
| **Practical competences:** | | | | | | |
| Correct use of principles and theoretical knowledge in practice | 4.13 | 4.34 | 4.03 | 3.61 | 3.94 | 3.44 |
| Use of information and technology (ICT) | 4.15 | 4.35 | 3.95 | 3.40 | 3.85 | 3.29 |
| Use of mathematics to solve operational problems | 3.06 | 3.69 | 2.88 | 3.39 | 3.00 | 3.36 |
| Understanding the business reality | 4.11 | 4.41 | 3.98 | 3.86 | 3.98 | 3.71 |

Note: Scales: Skills and competences needed to build global competences as perceived by students: 1 = to a very small extent and 5 = to a very large extent; The university provides the acquisition of the competence: 1 = to a very small extent and 5 = to a very large extent. Source: Authors' own research.

**Table 4.** Universities' activities that contribute to skills' development for the national versus international labor market as seen by students.

| Activities Related to the Internationalization of University | To What Extent Do the Activities of Your University Contribute to the Development of the Skills for the National Labor Market? | | To What Extent Do the Activities of Your University Contribute to the Development of Skills for the International Labor Market? | |
|---|---|---|---|---|
| | **NUPSPA—CPR** | **BUES—IBE** | **NUPSPA—CPR** | **BUES—IBE** |
| The curricula include intensive English courses/or subjects taught exclusively in English (A1) | 3.56 | 4.08 | 3.56 | 4.18 |
| The curricula include intensive courses in other foreign languages/or subjects taught exclusively in other languages (A2) | 2.71 | 3.44 | 2.96 | 3.64 |
| The curricula offer joint/dual diplomas from different countries (A3) | 3.32 | 3.19 | 3.45 | 3.35 |
| The curricula have an international topic preparing students for international professions (A4) | 3.55 | 3.74 | 3.61 | 3.76 |
| The curricula include compulsory subjects studied in institutions outside the country (A5) | 3.65 | 3.50 | 3.65 | 3.59 |
| The curricula lead to internationally recognized professional qualifications (A6) | 3.84 | 3.69 | 3.74 | 3.84 |
| Extracurricular activities promoted by the university for interconnection with groups belonging to local and international communities (A7) | 3.86 | 3.51 | 3.66 | 3.59 |
| Mobility of study programs of twinning type, double or joint specialization (A8) | 3.42 | 3.24 | 3.47 | 3.41 |
| Student mobility (A9) | 3.94 | 3.99 | 3.91 | 3.96 |
| University professors and staff mobility (A10) | 3.86 | 3.32 | 3.82 | 3.39 |
| University is attracting students, researchers, and developers of foreign educational programs as well as foreign companies in order to develop new programs and to provide new skills (A11) | 3.89 | 3.35 | 3.76 | 3.58 |

Note: Scales: The university contributes by its activities to the development of the skills for the national/international labor market: 1 = to a very small extent and 5 = to a very large extent. Source: Authors' own research.

Information in Table 4 and associated comments answer the research questions: RQ4 (What are the university's activities that contribute to the development of the skills for the national labor market?) and RQ5 (What are the university's activities that contribute to the development of the skills for the international labor market?).

### 4.2. Correlations between Skills and University's/Faculty's Activities

The mean calculation provides a general view of groups' tendency. In order to identify the type of correlation that exists between the skills and abilities provided by the two universities for the national/international labor market and the activities developed in universities, it is of interest to calculate the Pearson correlation between all 23 items (skills and abilities) that compose the global competences and the 11 types of activities of the two universities (A1–A11) perceived to contribute to the development of these skills. The Pearson correlation is generally interpreted as a "measurement of how much of the time changes in one variable correspond with equivalent changes in other variables" in order to understand the relation between two variables [54]. Since the two sample groups are larger than 100 subjects, we used the Evans empirical classifications of interpreting correlation strength by using r. The value of r is interpreted as being weak (0.20–0.39), moderate (0.40–0.59), strong (0.60–0.79), and very strong (0.80–1), when the *p* value is lower than 0.05 [57].

Overall, it can be stated that there is weak to moderate correlation between the provided competences and the university's/faculty's activities for the two analyzed groups, as the correlation coefficients vary from r = 0.13 to r = 0.62. Based on this, it can be stated that the activities carried out in the two universities/faculties contribute to the development of the skills analyzed. However, it can be observed that there are some notable differences between the correlation of the analyzed competences (with their associated skills) and the activities developed by the university/faculty depending on group of students and on national versus international labor markets. There are also activities that are correlated to a higher extent to the skills and abilities developed by the universities/faculties than other activities. The differences in the level of correlation are encountered between different university/faculty activities, different students' groups (the two specializations), and different labor markets (national and international) and are presented in the Supplementary Materials. Table 5 presents the Pearson correlation coefficients between all 23 skills and abilities considered by students to have been developed through their study programs and the first three universities'/faculties' activities that registered the highest correlation coefficients overall and are also statistically significant.

The results illustrate some of the differences:

(1) For the *national labor market*, there are three activities that present moderate correlation with the analyzed skills and abilities: A1, A10, and A11 as observed in Table 5. However, there are some differences between the two groups. In the case of BUES—IBE, there is an almost strong correlation between curricula that include intensive English courses/or subjects taught exclusively in English (A1) and the thorough knowledge of English language as it is provided by university (r = 0.56). Additionally, the implication of University in attracting students, researchers, and developers of foreign educational programs, as well as foreign companies in order to develop new programs and to provide new skills (A11), finds a notable correlation with competences like Organization and Problem solving (r = 0.54), Orientation towards customers and results (r = 0.51), Use of mathematics to solve operational problems (r = 0.52), and Understanding the business reality (r = 0.49).

**Table 5.** Correlation of university activities with global competences obtained for the NATIONAL labor market.

| Skills and Abilities | Pearson Correlation: Skills and Abilities Ensured by University for the National Labor Market (SA1–SA23) with University Activities (A1, A10, A11) | | | | | |
|---|---|---|---|---|---|---|
| | **A1** | | **A10** | | **A11** | |
| | **NUPSPA—CPR** | **BUES—IBE** | **NUPSPA—CPR** | **BUES—IBE** | **NUPSPA—CPR** | **BUES—IBE** |
| **International competences:** | | | | | | |
| Ability to communicate and to work efficiently in teams (SA1) | 0.13 | 0.27 ** | 0.24 ** | 0.24 ** | 0.22 ** | 0.30 ** |
| Thorough knowledge of English language (SA2) | 0.39 ** | 0.56 ** | 0.37 ** | 0.31 ** | 0.40 ** | 0.39 ** |
| Thorough knowledge of a foreign language other than English (SA3) | 0.42 ** | 0.36 ** | 0.31 ** | 0.25 ** | 0.38 ** | 0.37 ** |
| Understanding the international political, economic, and business context (SA4) | 0.31 ** | 0.38 ** | 0.47 ** | 0.25 ** | 0.48 ** | 0.31 ** |
| Ability to work in multidisciplinary teams (SA5) | 0.23 ** | 0.21 ** | 0.41 ** | 0.37 ** | 0.37 ** | 0.48 ** |
| Ability to work in multicultural teams (SA6) | 0.41 ** | 0.26 ** | 0.42 ** | 0.35 ** | 0.44 ** | 0.44 ** |
| **Personal competences:** | | | | | | |
| Integrity (SA7) | 0.38 * | 0.41 ** | 0.43 ** | 0.41 ** | 0.46 ** | 0.44 ** |
| Adaptability (SA8) | 0.31 ** | 0.28* | 0.37 ** | 0.38 ** | 0.46 ** | 0.47 ** |
| Initiative and involvement (SA9) | 0.34 ** | 0.30 ** | 0.42 ** | 0.32 ** | 0.50 ** | 0.46 ** |
| **Competences related to career management:** | | | | | | |
| Learning about the labor market and employment opportunities (SA10) | 0.36 ** | 0.33 ** | 0.55 ** | 0.30 ** | 0.54 ** | 0.46 ** |
| Use of the Internet and recruitment platforms to find a job (SA11) | 0.44 ** | 0.19* | 0.44 ** | 0.26 ** | 0.46 ** | 0.43 ** |
| Ability to present and to produce a CV (SA12) | 0.30 ** | 0.28 ** | 0.38 ** | 0.34 ** | 0.45 ** | 0.46 ** |
| **Workplace competences:** | | | | | | |
| Planning (SA13) | 0.33 ** | 0.27 ** | 0.46 ** | 0.54 ** | 0.53 ** | 0.48 ** |
| Organization (SA14) | 0.36 ** | 0.32 ** | 0.44 ** | 0.48 ** | 0.51 ** | 0.54 ** |
| Problem solving (SA15) | 0.39 ** | 0.32 ** | 0.52 ** | 0.50 ** | 0.48 ** | 0.54 ** |
| Orientation towards customers and results (SA16) | 0.35 ** | 0.32 ** | 0.45 ** | 0.38 ** | 0.49 ** | 0.51 ** |
| Critical thinking in decision making process (SA17) | 0.28 ** | 0.31 ** | 0.33 ** | 0.33 ** | 0.36 ** | 0.39 ** |

**Table 5.** *Cont.*

| Skills and Abilities | Pearson Correlation: Skills and Abilities Ensured by University for the National Labor Market (SA1–SA23) with University Activities (A1, A10, A11) | | | | | |
|---|---|---|---|---|---|---|
| | **A1** | | **A10** | | **A11** | |
| | **NUPSPA—CPR** | **BUES—IBE** | **NUPSPA—CPR** | **BUES—IBE** | **NUPSPA—CPR** | **BUES—IBE** |
| **Theoretical competences:** | | | | | | |
| Knowledge of general economic theoretical concepts and formation of an economic way of thinking (SA18) | 0.24 ** | 0.33 ** | 0.42 ** | 0.18 * | 0.44 ** | 0.27 ** |
| Knowledge of professional terminology, theories, and concepts related to profession (SA19) | 0.36 ** | 0.23 ** | 0.49 ** | 0.23 ** | 0.45 ** | 0.28 ** |
| **Theoretical competences:** | | | | | | |
| Correct use of principles and theoretical knowledge in practice (SA20) | 0.40 ** | 0.36 ** | 0.53 ** | 0.38 ** | 0.50 ** | 0.47 ** |
| Use of information and technology (ICT) (SA21) | 0.47 ** | 0.21* | 0.43 ** | 0.41 ** | 0.47 ** | 0.47 ** |
| Use of mathematics to solve operational problems (SA22) | 0.36 ** | 0.23 ** | 0.37 ** | 0.38 ** | 0.45 ** | 0.52 ** |
| Understanding the business reality (SA23) | 0.40 ** | 0.30 ** | 0.55 ** | 0.37 ** | 0.56 ** | 0.49 ** |

Note: *. Correlation is significant at the 0.05 level (2-tailed); **. Correlation is significant at the 0.01 level (2-tailed). Source: Authors' own research, valuing SPSS output. The values are rounded at two decimal places.

In the case of NUPSPA—CPR, there is a moderate correlation between the Implication of university in attracting students, researchers, and developers of foreign educational programs as well as foreign companies in order to develop new programs (A11) and the global competences like Organization (r = 0.50), Problem solving (r = 0.51), Orientation towards customers and results (r = 0.49), Understanding the business reality (r = 0.56), Correct use of principles and theoretical knowledge in practice (r = 0.49), Planning (r = 0.53), Learning about the labor market and employment opportunities (r = 0.54), and Initiative and involvement (r = 0.50). There is also a moderate correlation between the Mobility of professors and staff (A10) and the development of the practical competence like Correct use of principles and theoretical knowledge in practice (r = 0.51), Understanding the business reality (r = 0.55), workplace competences like Problem solving (r = 0.52), or competences related to career management like Learning about the labor market and employment opportunities (r = 0.55).

Both student categories from BUES-IBE and NUPSPA-CPR consider that a significant part of the whole package of "workplace skills" (such as planning, organization, and problem solving) are provided by their own universities through the A11 activity. In other words, Attracting researchers and specialists in the field of research projects of universities (A11) has a direct impact on increasing the "workplace skills" of students.

(2) For the *international labor market*, the analyses revealed five activities (out of eleven) that involve moderate correlation with skills and abilities provided by the two universities as seen in Table 6. In case of BUES—IBE, there is a moderate correlation between the International topic of curricula preparing students for international professions (A4) and the global competences like: Ability to communicate and to work efficiently in teams (r = 0.52), Understanding the international political, economic, and business context (r = 0.50), Adaptability and planning (r = 0.49), Organization (r = 0.57), Problem solving (r = 0.52), and Knowledge of professional terminology, theories, and concepts related to profession (r = 0.50). A moderate correlation was present as well between the Mobility of professors and staff (A10) and competences and associated skills such as: Initiative and involvement (r = 0.53), Learning about the labor market and employment opportunities (r = 0.51), Organization (r = 0.54), Problem solving (r = 0.50), Correct use of principles and theoretical knowledge in practice (r = 0.53), and Use of mathematics to solve operational problems and understanding the business reality (r = 0.51). Furthermore, a moderate correlation was present between the activity of Attracting students, researchers, and developers of foreign educational programs as well as foreign companies in order to develop new programs (A11) and the competences like Learning about the labor market and employment opportunities (r = 0.54), Use of mathematics to solve operational problems and understanding the business reality (r = 0.51).

The analysis for the NUPSPA—CPR revealed a moderate correlation between all selected five activities (the ones that have the highest correlation coefficients) and almost all the skills and abilities that composed the global competences revealing the score ranges, detailed in the Table 6. The strongest correlation is encountered for the relationship between Extracurricular activities promoted by the university for interconnection with groups belonging to local and international communities (A7) and skills such as Planning (r = 0.62), Understanding the international political, economic, and business context (r = 0.61), and Orientation towards customers and results (r = 0.61).

There are notable differences between the correlation of university activities and the skills and abilities provided for the national vs. international labor market. Thus, the correlation shows higher values for the international labor market for both universities. The five activities that show higher correlation with skills and abilities provided for the international labor market are: Internationalization of curricula (A1—curricula include intensive English courses/or subjects taught exclusively in English and A4—curricula have an international topic preparing students for international professions); extracurricular activities promoted by the university for interconnection with groups belonging to local and international communities (A7) and activities focused on cross border education (A10 and A11—mobilities of teachers, researchers, and specialists from international universities or companies).

**Table 6.** Correlation of university activities with global competences obtained for the International labor market.

| Skills and Abilities | Pearson Correlation Skills and Abilities Ensured by University for the International Labor Market (SA1–SA23) with University Activities (A1, A4, A7, A10, A11) | | | | | | | | | |
|---|---|---|---|---|---|---|---|---|---|---|
| | A1 | | A4 | | A7 | | A10 | | A11 | |
| | NUPSPA—CPR | BUES—IBE | NUPSPA—CPR | BUES—IBE | NUPSPA—CPR | BUES—IBE | NUPSPA—CPR | BUES—IBE | NUPSPA—CPR | BUES—IBE |
| SA1 | 0.35 ** | 0.37 ** | 0.29 ** | 0.52 ** | 0.40 ** | 0.34 ** | 0.50 ** | 0.38 ** | 0.41 ** | 0.36 ** |
| SA2 | 0.60 ** | 0.49 ** | 0.45 ** | 0.46 ** | 0.50 ** | 0.37 ** | 0.47 ** | 0.33 ** | 0.42 ** | 0.33 ** |
| SA3 | 0.53 ** | 0.34 ** | 0.50 ** | 0.33 ** | 0.46 ** | 0.31 ** | 0.35 ** | 0.36 ** | 0.43 ** | 0.30 ** |
| SA4 | 0.41 ** | 0.38 ** | 0.48 ** | 0.50 ** | 0.61 ** | 0.30 ** | 0.56 ** | 0.38 ** | 0.53 ** | 0.33 ** |
| SA5 | 0.47 ** | 0.22 ** | 0.48 ** | 0.38 ** | 0.58 ** | 0.44 ** | 0.59 ** | 0.48 ** | 0.54 ** | 0.47 ** |
| SA6 | 0.50 ** | 0.37 ** | 0.44 ** | 0.32 ** | 0.64 ** | 0.42 ** | 0.55 ** | 0.42 ** | 0.59 ** | 0.36 ** |
| SA7 | 0.45 * | 0.43 ** | 0.38 ** | 0.36 ** | 0.56 ** | 0.36 ** | 0.58 ** | 0.42 ** | 0.50 ** | 0.35 ** |
| SA8 | 0.42 ** | 0.32 * | 0.32 ** | 0.49 ** | 0.49 ** | 0.38 ** | 0.54 ** | 0.47 ** | 0.45 ** | 0.37 ** |
| SA9 | 0.35 ** | 0.32 ** | 0.37 ** | 0.47 ** | 0.46 ** | 0.48 ** | 0.47 ** | 0.53 ** | 0.43 ** | 0.45 ** |
| SA10 | 0.42 ** | 0.34 ** | 0.45 ** | 0.35 ** | 0.59 ** | 0.41 ** | 0.60 ** | 0.51 ** | 0.53 ** | 0.54 ** |
| SA11 | 0.58 ** | 0.33 ** | 0.59 ** | 0.33 ** | 0.58 ** | 0.47 ** | 0.57 ** | 0.46 ** | 0.53 ** | 0.48 ** |
| SA12 | 0.45 ** | 0.30 ** | 0.47 ** | 0.34 ** | 0.50 ** | 0.34 ** | 0.43 ** | 0.47 ** | 0.43 ** | 0.43 ** |
| SA13 | 0.43 ** | 0.26 ** | 0.51 ** | 0.49 ** | 0.62 ** | 0.42 ** | 0.61 ** | 0.49 ** | 0.58 ** | 0.44 ** |
| SA14 | 0.46 ** | 0.26 ** | 0.44 ** | 0.57 ** | 0.57 ** | 0.40 ** | 0.60 ** | 0.54 ** | 0.49 ** | 0.47 ** |
| SA15 | 0.49 ** | 0.32 ** | 0.47 ** | 0.53 ** | 0.56 ** | 0.35 ** | 0.57 ** | 0.50 ** | 0.53 ** | 0.40 ** |
| SA16 | 0.40 ** | 0.28 ** | 0.48 ** | 0.40 ** | 0.61 ** | 0.29 ** | 0.59 ** | 0.40 ** | 0.60 ** | 0.43 ** |
| SA17 | 0.34 ** | 0.33 ** | 0.31 ** | 0.45 ** | 0.40 ** | 0.27 ** | 0.44 ** | 0.48 ** | 0.35 ** | 0.39 ** |
| SA18 | 0.40 ** | 0.37 ** | 0.41 ** | 0.48 | 0.54 ** | 0.16 | 0.52 ** | 0.31 ** | 0.53 ** | 0.26 ** |
| SA19 | 0.43 ** | 0.34 ** | 0.47 ** | 0.50 ** | 0.56 ** | 0.25 ** | 0.54 ** | 0.34 ** | 0.55 ** | 0.32 ** |
| SA20 | 0.38 ** | 0.41 ** | 0.47 ** | 0.40 ** | 0.55 ** | 0.30 ** | 0.57 ** | 0.53 ** | 0.59 ** | 0.45 ** |
| SA21 | 0.54 ** | 0.32 ** | 0.39 ** | 0.36 * | 0.56 ** | 0.28 ** | 0.52 ** | 0.42 ** | 0.52 ** | 0.44 ** |
| SA22 | 0.46 ** | 0.25 ** | 0.47 ** | 0.47 ** | 0.46 ** | 0.35 ** | 0.34 ** | 0.51 ** | 0.39 ** | 0.51 ** |
| SA23 | 0.42 ** | 0.27 ** | 0.46 ** | 0.44 ** | 0.55 ** | 0.43 ** | 0.54 ** | 0.48 ** | 0.51 ** | 0.52 ** |

Note: *. Correlation is significant at the 0.05 level (2-tailed); **. Correlation is significant at the 0.01 level (2-tailed). Source: Authors' own research, valuing SPSS output. The values are rounded at two decimal places.

Based on the study of the correlation between skills and abilities development and the university's/faculty's activities contributing to their development, it can be observed that the BUES-IBE program, specialized in international economic relations, is perceived to be strong in foreign languages and foreign curricula, while the NUSPA-CPR program, specialized in social and communication studies, is strong in the extra-curricular activities, while both programs are perceived to be good in attracting students, researchers, and developers of foreign educational programs as well as foreign companies in order to develop new programs and to provide new skills.

It can be concluded that Hypothesis 4 was only partially verified, as the correlations between skills developed by study programs and the university's/faculty's activities are weak to moderate and are not moderate to strong as hypothesized.

Information in Tables 5 and 6 and the associated comments answer the research question RQ6 (How do the skills and abilities perceived by students as being ensured by their university/faculty at a general level correlate to the contributions that specific activities conducted in the university/faculty have to the development of skills and abilities?).

### 4.3. Testing Significant Differences Based on t-*test between Students' Perceptions in the Two Universities/Faculties*

In addition to the analysis of the Pearson correlations presented above, it is of interest to evaluate whether the means of the two independent groups of students differ on some variables. Mean difference *t*-test is used for this purpose. The independent samples *t*-test compares the means of two independent groups in order to determine whether there is statistical evidence that the associated population means are significantly different [35].

To test whether there are significant differences between the two independent groups (students from NUSPA-CPR program and students from BUES-IBE program), we performed the hypotheses testing for differences in means between the two independent samples.

$H_0 : \mu_1 - \mu_2 = 0$ (there are no significant differences between the two independent groups)

$H_1 : \mu_1 - \mu_2 \neq 0$ (there are significant differences between the two independent groups)

The level of confidence is set at 95%, $\alpha = 0.05$ and for *p*-value < 0.05, reject H0.

The hypotheses testing is run for the 3 different approaches to the global competences and their associated skills: (a) The students' perception of the contribution of a number of skills to the development of global competences (Hypothesis 1); (b) the students' perception of the extent to which the university/faculty they attend contributes to the development of the global competences (and associated skills) necessary for the national labor market (Hypothesis 2); and (c) the students' perception of the extent to which the university/faculty they attend contributes to the development of the global competences (and associated skills) necessary for the international labor market (Hypothesis 3). In *t*-test approach, the most relevant difference between the averages is at $p = 0.000$ level of significance (2-tailled) and statistically significant differences exist at *p*-value < 0.05. The areas marked with greys in Table 7 represent significant differences between the averages of statistical classes.

From the results presented in Table 7, since for the majority of variables the *p* values are lower than 0.05, the null hypotheses will be rejected and it can be concluded, as a general appreciation, that there are statistically significant differences between the means of the two groups of students.

**Table 7.** Testing significant differences between student groups based on *t*-test.

| Global Competences and Associated Skills and Abilities | Skills and Abilities Needed to Obtain Global Competences (Hypothesis 1) | | The University Ensures the Development of the Skills Needed to Activate in the NATIONAL Labor Market (Hypothesis 2) | | The University Ensures the Development of the Skills Needed to Activate in the INTERNATIONAL Labor Market (Hypothesis 3) | |
|---|---|---|---|---|---|---|
| | t Value | Sig. (2-Tailed) *p*-Value | t Value | Sig. (2-Tailed) *p*-Value | t Value | Sig. (2-Tailed) *p*-Value |
| **International competences:** | | | | | | |
| Ability to communicate and to work efficiently in teams | 0.54 | 0.592 | 3.87 | 0.000 | 2.77 | 0.006 |
| Thorough knowledge of English language | −2.88 | 0.004 | −4.40 | 0.000 | −2.94 | 0.004 |
| Thorough knowledge of a foreign language other than English | −7.63 | 0.000 | −5.15 | 0.000 | −3.96 | 0.000 |
| Understanding the international political, economic, and business context | −6.59 | 0.000 | −2.17 | 0.031 | −1.08 | 0.28 |
| Ability to work in multidisciplinary teams | −3.12 | 0.002 | 3.85 | 0.000 | 3.49 | 0.001 |
| Ability to work in multicultural teams | −1.34 | 0.181 | 1.20 | 0.230 | 2.19 | 0.029 |
| **Personal competences:** | | | | | | |
| Integrity | 1.24 | 0.216 | 3.84 | 0.000 | 3.56 | 0.000 |
| Adaptability | 0.63 | 0.528 | 3.61 | 0.000 | 3.34 | 0.001 |
| Initiative and involvement | 0.61 | 0.540 | 3.89 | 0.000 | 3.72 | 0.000 |
| **Competences related to career management:** | | | | | | |
| Learning about the labor market and employment opportunities | −2.94 | 0.004 | 2.97 | 0.003 | 5.28 | 0.000 |
| Use of the Internet and recruitment platforms to find a job | −1.13 | 0.259 | 2.12 | 0.035 | 3.59 | 0.000 |

**Table 7.** *Cont.*

| Global Competences and Associated Skills and Abilities | Skills and Abilities Needed to Obtain Global Competences (Hypothesis 1) | | The University Ensures the Development of the Skills Needed to Activate in the NATIONAL Labor Market (Hypothesis 2) | | The University Ensures the Development of the Skills Needed to Activate in the INTERNATIONAL Labor Market (Hypothesis 3) | |
|---|---|---|---|---|---|---|
| | t Value | Sig. (2-Tailed) *p*-Value | t Value | Sig. (2-Tailed) *p*-Value | t Value | Sig. (2-Tailed) *p*-Value |
| Ability to present and to produce a CV | 0.89 | 0.372 | 2.24 | 0.026 | 3.57 | 0.000 |
| **Workplace competences:** | | | | | | |
| Planning | 0.22 | 0.826 | 3.71 | 0.000 | 4.3 | 0.000 |
| Organization | 0.44 | 0.658 | 3.75 | 0.000 | 4.59 | 0.000 |
| Problem solving | 0.23 | 0.817 | 3.18 | 0.002 | 4.05 | 0.000 |
| Orientation towards customers and results | −0.12 | 0.902 | 3.99 | 0.000 | 3.85 | 0.000 |
| Critical thinking in decision making process | −0.50 | 0.617 | 3.77 | 0.000 | 3.57 | 0.000 |
| **Theoretical competences:** | | | | | | |
| Knowledge of general economic theoretical concepts and formation of an economic way of thinking | −4.97 | 0.000 | −2.06 | 0.041 | −0.80 | 0.423 |
| Knowledge of professional terminology, theories, and concepts related to profession | −3.11 | 0.002 | −0.40 | 0.687 | −0.45 | 0.652 |
| **Practical competences:** | | | | | | |
| Correct use of principles and theoretical knowledge in practice | −2.17 | 0.031 | 3.50 | 0.001 | 4.06 | 0.000 |
| Use of information and technology (ICT) | −1.99 | 0.048 | 4.33 | 0.000 | 4.11 | 0.000 |
| Use of mathematics to solve operational problems | −5.43 | 0.000 | −3.66 | 0.000 | −2.50 | 0.013 |
| Understanding the business reality | −3.10 | 0.002 | 1.04 | 0.119 | 2.29 | 0.023 |

Source: Authors' own research valuing SPSS output.

Looking at the students' perception of the contribution of a number of skills to the development of global competences (Hypothesis 1), it can be observed that there are statistically significant differences for the International Competences, Theoretical Competences, and Practical Competences. This means that students from the two specializations (economic and social) have different opinions about which skills contribute to the development of the above-mentioned global competences categories. However, as far as the Personal Competences, Competences Related to the Career Management, and Work-place Competences, the two groups of students have rather similar opinions, as the differences are not statistically significant. One possible explanation relates to the fact that students in both universities/faculties have little or no job experience and therefore, they assume in a similar manner what might be necessary for career management and they have similar perceptions related to their expected behavior at the work place. It can be concluded that Hypothesis 1 (The students' perception of "skills and abilities needed to obtain global competencies" presents statistically significant differences between the two universities/faculties with different specializations) is partially verified.

Looking at the perception of students on the extent to which the university/faculty they attend contributes to the development of the global competences (and associated skills) necessary for the national labor market (Hypothesis 2) and for the international labor market (Hypothesis 3), it can be noticed from Table 7 that most of the *p*-values are lower than 0.05, therefore illustrating statistically significant differences between the students with the two specializations for the extent to which the university/faculty they attend ensures the development of the skills needed to activate in the labor market. There is only one skill (Knowledge of professional terminology, theories, and concepts related to profession) for which there were no statistically significant differences between the two groups of students in the case of both national and international labor markets, and some other three skills (see Table 7) with no significant differences in the perceptions of the students from the two specializations, for the extent to which their university/faculty ensures the development of those skills, for either the national labor market or the international labor market. Therefore, it can be concluded that both Hypotheses 2 and 3 have been verified.

Information in Table 7 and associated comments answer the research question RQ7 (Are there any differences between opinions of students from the two different faculties with different specializations (from the two universities)?).

## 5. Discussion

The present study reveals a number of aspects connected to global competences that relate to existing literature and other studies' findings:

(1) In this study, for the two independent groups of students from universities with different fields of study, there are common opinions regarding the skills and abilities needed to obtain global competences on the componence of Personal competences, Competences related to career management, Workplace competences, and some International competences. Thus, both groups of students are equally appreciating skills (as contributors to competences development) like: Integrity, Adaptability, Initiative and involvement, Use of Internet and recruitment platforms to find a job, Ability to present and to produce a CV, Planning, Organization, Problem solving, Orientation towards customers and results, Critical thinking, Ability to communicate and to work efficiently in teams, and the Ability to work in multicultural teams. The results of the present study resemble the findings of other studies that illustrate how competences such as communication skills, initiative, business ethics, foreign language ability, overall learning ability, adaptive ability, and self-control are equally important for the international business graduates [58] as seen by company executives [26]. Similarly, other authors [38,59,60] found that skills such as personal flexibility, ambition, trustworthiness, reliability, motivation, communication skills, ability to work under pressure, communication with customers/clients, interpersonal communication with co-workers, oral presentation skills, and a willingness to learn are important for economics, management, and marketing graduates and also for social and humanities graduates [30,61,62] as seen by both graduates and companies.

(2) The are differences between the two groups on the level of perceived needed skills and abilities, and they refer to the skills and abilities that are related to international competences. Skills and abilities like Knowledge of English language and other foreign languages, Understanding the international political, economic, and business context, and Ability to work in multidisciplinary teams are perceived as being more important by students from the BUES-IBE. This shows that for the international business economic students it is essential for their future career to develop skills and abilities that will offer them the connection with the international markets and international business realities [26,58], and this happens to a larger extent than in the case of students specialized in social studies. Similarly, a previous study looking at the student experiences in the same two higher education institutions identified differences in the perception of students of their study programs [63].

(3) Additionally, this study revealed that in general, that Romanian universities are perceived as offering the needed theoretical knowledge for the graduates, as averages above 3.5 were obtained for the questions related to the acquisition of theoretical knowledge, and no statistically significant differences between the two groups of students were encountered. This comes to reinforce previous findings on Romania [22,64] that place the theoretical knowledge as one strength of higher education institutions in Romania, from both students' and employers' perspectives.

(4) For the other skills and abilities' categories that composed the global competences, there are different perceptions of students from the two specializations as regards the extent to which such skills are provided by their university/faculty. Thus, on the one hand, the students from BUES—IBE consider that their university is more oriented towards providing international and specific competences like Knowledge of English language, Use of mathematics to solve operational problems, or Understanding the international, political, economic, and business context, while on the other hand, the NUPSPA—CPR students are considering, to a large extent, that competences like Ability to work in multidisciplinary teams, Orientation towards customers and results, and Critical thinking in decision making process are better ensured by their university, reconfirming differences in opinions of students with different specializations [63]. The differences in perceptions (between the two groups) are similar for the competences provided by the university/faculty to activate in the national and also in the international labor market.

(5) There is a connection between universities' activities and many of the provided global competences for both the national and international labor markets. This can be seen as a consequence of the Romanian universities' efforts, in the later years, to internationalize the educational process, as part as their new managerial strategies [65], by applying internationalization strategies in HE institutions in order to improve the quality of education by opening to international markets [18]. The internationalization of HE in Romania aims to prepare the students to become active socially responsible citizens able to cope with all aspects of life in a global society [12,18]. Prior to the implementation of the internationalization strategies at HE institution level, the studies have shown that students' mobility [66] was the main form of internationalization in Romania [67], ensuring access to international competences for only a small number of students [68]. However, the present study illustrates that students recognize furthermore other activities that are connected with the skills and abilities provided by universities for the national and international labor market.

This is more prevalent in the case of BUES—IBE, where the moderate correlation between activities like internationalization of curricula (Curricula include intensive English courses/or subjects taught exclusively in English and Curricula have an international topic preparing students for international profession) [16] and specific international, theoretical, and workplace skills and abilities like Ability to communicate and to work efficiently in teams, Thorough knowledge of English language, Knowledge of professional terminology, theories and concepts related to profession, Problem solving, and Organization, depicts the opinions of students that the internationalization of curricula leads to global competences, as acknowledged by the literature and other studies [16,30,39,68,69]. However, for both universities, there is a correlation between the activities focused on cross-border education (mobility of professors, students, and researchers to develop new programs and research

projects), seen as important drivers in the internationalization activity of a university [12,16,39,69], and the personal, practical, and workplace competences like: Adaptability, Initiative and involvement, Planning, Organization, Problem solving, Orientation towards customers and results, and Correct use of principles and theoretical knowledge in practice [43].

(6) The contribution of different universities' activities to the competences and skills development is subject-related, as in the case of NUPSPA—CPR (social studies specialization), a higher correlation is present between the extracurricular activities and the skills and competences belonging to Personal, Practical, and Workplace competences, different in comparison with the economic and international economic relations specialization, where the connection with the internationally-oriented activities was stronger. This needs to be considered by universities' management [65] when deciding on which types of activities to focus on more in order to ensure the development of skills and competences needed in the case of each specialization, in the conditions of ensuring a qualitative educational process [64].

## 6. Conclusions, Limitations, and Future Research

The present study looked at students' opinions from two specializations (social studies and economic studies) on global competences development, reflecting certain commonalities and differences between the two. The theoretical framework for the study of global competences was developed based on the existing literature related to skills, abilities, and competences, and resulted in a typology of global competences that comprised six major global competences (international competences, personal competences, competences related to career management, workplace competences, theoretical competences, and practical competences), each of them being built from a number of skills and abilities. All of them have an influence on employability according to the literature [10,29,33,36,37], and the present study also confirms that students consider all six categories of global competences as being necessary to activate in both national and international labor markets.

More specifically, it was observed that students studying in social studies were inclined to consider personal competences as being the most needed to build global competences, while students studying in economic fields (specialization, international business, and economics) consider international competences as being the most important to build global competences. In the same vein, university/ faculty activities oriented towards internationalization (both at home and abroad) were seen as being the main contributors to the development of skills needed in both national and international labor markets by students who study in the economic field. At the same time, extra-curricular activities and business and academia collaborations, among university's/faculty's activities, were seen by students who study in the social field, as main contributors to the development of global competences required in labor markets.

Additionally, the two groups of students presented statistically significant differences in their opinions about certain categories of global competences and included skills and abilities (International Competences, Theoretical Competences, and Practical Competences), illustrating that they have rather different opinions about the contribution of different types of skills to the development of global competences in general, but also on the contribution of their respective university/faculty to the development of these categories of skills, abilities, and competences. Other categories of competences (Personal Competences, Competences Related to the Career Management, and Work-Place Competences) seem more similar in the eyes of the two groups of students.

It can be concluded that in general, students have well delineated opinions on the skills needed to build global competences, while there are some differences coming from different fields of studies in respect to which skills will contribute to which competence development in the context in which universities/faculties themselves are perceived as offering different levels of skill development through their activities, depending on the field of study and on the type of labor market envisaged (national or international).

The implications of the results of the present study are multifold: (a) For higher education authorities and policy makers, based on the results of the present study, it is recommended to support

the development of a stronger international character of universities, faculties, and study programs in order to develop global competences also required on international labor markets; (b) for higher education institutions one main direct implication at organizational level would be the development of more collaborative study programs that would imply international partners both from the academia and the business environment, as appreciated by students, and (c) at student level, one implication would be that they need to develop a combination of skills, abilities, and competences in order to increase their employability in both national and international labor markets. The results of the study would imply the need for redefining the curricula by including more international components, and this is a recommendation at higher education institution level, that it can also be encouraged through educational policy measures via designated programs.

As any research study, this study has also limitations. Among those, we can mention the inclusion of only two fields of studies (economic and social studies). Additionally, the fact that the study has been conducted in a specific national environment might affect results that also might be culturally related and not entirely generalizable at a larger level.

In the context of the existing limitations, we forward some suggestions for future research. For a more in-depth analysis, a confirmatory research could be conducted on the potential correlation between the ideal skills and abilities needed to build global competences, as perceived by students and the actual skills and abilities offered by different faculties and study programs. Additionally, in order to build a more comprehensive image about the development of global competences during the educational process in higher education institutions of diverse profiles, the study can be extended through further research by including other fields of study (such as technical studies, medical studies, and others) and identify a broader range of field related global competences. Another research direction can be the development of a cross-national sample that would also include students from different countries in order to identify potential differences in various national context and also identify and reinforce the common grounds at international level.

**Supplementary Materials:** The following are available online at http://www.mdpi.com/2071-1050/12/13/5364/s1. Table S1: Correlation of university activities with global competences obtained for the national labor market for the two institutions, Table S2: Correlation of university activities with global competences obtained for the international labor market for the two institutions.

**Author Contributions:** Conceptualization, L.C.B. and L.N.; data curation, L.C.B., L.N., S.O.S., and A.G.; formal analysis, L.C.B., L.N., and A.G.; investigation, L.C.B., L.N., S.O.S., and A.G.; methodology, L.C.B., L.N., and S.O.S.; project administration, L.C.B.; resources, L.C.B. and L.N.; software, L.C.B., S.O.S., and A.G.; supervision, L.C.B. and L.N.; visualization, L.C.B., L.N., S.O.S., and A.G.; writing—original draft preparation, L.C.B.; writing—review and editing, L.C.B., L.N., S.O.S., and A.G.; funding acquisition. All authors have read and agreed to the published version of the manuscript.

**Funding:** This research received no external funding.

**Conflicts of Interest:** The authors declare no conflict of interest.

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
