# Peer review of "Providing Sustainable Knowledge for the Young Graduates of Economic and Social Sciences. Case Study: Comparative Analysis of Required Global Competences in Two Romanian Universities"

_sustainability, doi:10.3390/su12135364_

Round 1

Reviewer 1 Report

The research is well conducted, scientific writing is appropriate. 

The introduction presents a good literature review, however, the most recent literature about skills for international skills are missing, and this is a very dynamic field, so you should consider research from 2018-2020.

Also, you should explain better the constructs studied, as the concept of skills is very controversial.

In the methodology, a section with the theorization of the methods and statistical techniques used must be included (as a 1st topic).

The results are well explained, but you should explain better the validity and reliability of the questionnaire.

Finally, the conclusions need to be specific and linked to the results and the theoretical framework.

Important also are the implications of the study for the educational actors involved, and also for the redefinition of the curricula and to give insights for policymakers. 

Congratulations on your study.

Author Response

Point 1: The introduction presents a good literature review, however, the most recent literature about skills for international skills are missing, and this is a very dynamic field, so you should consider research from 2018-2020.

Response 1: Thank you for your remarks. The paper was enriched with new studies both in the Theoretical framework section and the Discussion section.

The new studies and articles that have been consulted and cited are marked with red colour in the revised paper as it follows: 29, 32-37, 40-41, 46-56.

  1. Succi, C,; Canovi, M, Soft skills to enhance graduate employability: comparing students and employers’ perceptions. Studies in Higher Education 2019, DOI: 1080/03075079.2019.1585420, 1-14.
  2. Cleveland, S,; Cleveland, M, Leadership Competencies for Sustained Project Success. International Journal of Applied Management Theory and Research 2020, 2(1), 35-47.
  3. O’Leary, S, Graduates’ experiences of, and attitudes towards, the inclusion of employability-related support in undergraduate degree programmes; trends and variations by subject discipline and gender. Journal of Education and Work 2016, 30:1, 84-105.
  4. European Commission, Supporting Key Competences Development: Learning approaches and environments in school education. Conference Report, Brussels 12-13 November 2019. Available online: https://op.europa.eu/en/publication-detail/-/publication/37cc844f-4248-11ea-9099-01aa75ed71a1/language-en/format-PDF/source-search, (accessed 24 June 2020).
  5. Yari, N,; Lankut, E,; Alon, I,; Richter, N,F, Cultural Intelligence, global mindset, and cross-cultural competences: a systematic review using bibliometric methods. European J. International Mananagement 2020, 14(2), 210-250.
  6. Gedye, S,; Beaumont, E, The ability to get a job: student understandings and definitions of employability. Education + Training 2018, 60(5), 406-420.
  7. Monteiro, S,; Almeida, L,; Gomes, C,; Sinval, J, Employability profiles of higher education graduates: a person-oriented approach. Studies in Higher Education 2020, DOI: 1080/03075079.2020.1761785, Available online: http://repositorium.sdum.uminho.pt/handle/1822/65456 (accessed on 24 June 2020).
  8. Yorke, M,; Knight, P,T, Embedding employability into the curriculum. Learning and Employability Series I, The Higher Education Academy 2006. Available online: https://www.qualityresearchinternational.com/esecttools/esectpubs/yorkeknightembedding.pdf (accessed on 2 December 2018).
  9. Chhinzer, N,; Russo, A,M, An exploration of employer perceptions of graduate student employability. Education + Training 2018, 60(1), 104-120.
  10. Ala-Louko, R, Developing International Competence and Intercultural Communication Skills is an Investment in the Future. Lapland University of Applied Science, 2017, Available online: URN:NBN:fi:amk-2017092815485, (accessed on 23 June 2020).
  11. Singh, R,; Chawla, G,; Agarwal, S.; Desai, A, Employability and innovation: development of a scale. International Journal of Innovation Science 2017, 9(1), 20-37.
  12. Römgens, I,; Scoupe, R,; Beausaert, S, Unraveling the concept of employability, bringing together research on employability in higher education and the workplace. Studies in Higher Education 2019, DOI: 10.1080/03075079.2019.1623770, 1-16, Available online: https://www.tandfonline.com/doi/pdf/10.1080/03075079.2019.1623770, (accessed on 23 June 2020).
  13. Van der Heijden, B,I,J,M,; Notelaers, G,; Peters, P,; Stoffers, J,; De Lange, A,; Froehlich, D,E,; Van der Heijde, C,M, Development and validation of the short-form employability five-factor instrument. Journal of Vocational Behavior 2018, 106, 236-248.
  14. Caspersen, J,; Smeby, J-C, Placement training and learning outcomes in social work education. Studies in Higher Education 2020, DOI:10.1080/03075079.2020.1750583, 1-14, Available online: https://www.tandfonline.com/doi/pdf/10.1080/03075079.2020.1750583?needAccess=true (accessed on 25 June 2020)
  15. Gordon, L,E, Real research. Research Methods Sociology Students Can Use. SAGE Publications, Inc.: Thousand Oaks, USA, 2016, p.91.
  16. Creswell, J,W, Research Design. International Student Edition. Qualitative, Quantitative and Mixed Methods Approaches. SAGE Publications, Inc.: London, UK, 2014, p.163.
  17. Bolarinwa, O,A, Principles and methods of validity and reliability testing of questionnaires used in social and health science researches. Niger Postgrad Med J 2015, 22, 195-201
  18. Bernard, H,R, Social Research Methods. Qualitative and Quantitative Sage Publication Ltd.: Los Angeles, USA, 2013, p. 649.
  19. Taber, K,S, The Use of Cronbach’s Alpha When Developing and Reporting Research Instruments in Science Education. Res Sci Educ 2018, 48, 1273–1296
  20. Deniz, M,S,; Alsaffar, A,A, Assessing the Validity and Reliability of a Questionnaire on Dietary Fibre-related Knowledge in a Turkish Student Population. Journal of Health Population and Nutrition 2013, 31(4), 497-503.

Point 2: Also, you should explain better the constructs studied, as the concept of skills is very controversial.

Response 2: Were added to our study the following explanations (line 138-140 and 162-168):

Competences are seen in the literature from different perspectives: the students and/or graduates [22,36,37,38], the university [39,40] and the employers’ perspectives [22,33,41] and all of them have been considered when defining competences in the present study.

Both classical and recent studies on skills, abilities, competences and employability were used for this purpose as presented in Table 1. The study of the literature on skills, abilities and competences showed that their definitions and approaches are very diverse, sometimes controversial but they also present many commonalities. The perspective used to build the present typology was based on two guidelines: the inclusion of a large range of skills and abilities that contribute to individual employability and an extant agreement of the literature on what each category of competencies comprises.

Point 3: a) In the methodology, a section with the theorization of the methods and statistical techniques used must be included (as a 1st topic).

  1. b) The results are well explained, but you should explain better the validity and reliability of the questionnaire.

Response 3: a) The Research method and statistical techniques were presented distinctly, from line 175-189 as it follows:

3.1 Research method and statistical techniques

In the present research a survey was conducted, using a questionnaire designed to identify the students’ perceptions on global competences and on universities’/faculties’ activities for providing the competences needed on the national/international labour market. The survey was used as a data collection method as it allows a large number of responses and possible generalizations of results via statistical analysis [51] (p.91).

The sample was formed of two categories of students from two different study specializations, one that through its profile is oriented more towards internationalization and another that it is not. Besides identifying the perceptions on skills and competences for each group on its own, also a comparison between the two groups of students was envisaged.

One approach to statistical analysis consisted in descriptive statistics, used to measure the central tendencies [52] (p.163) of the two groups of students. Another approach to statistical analysis referred to inferential statistics that included correlations on the one hand and t-test for group differences, on the other hand, as a method that allows for group comparison [51,52]. The SPSS statistical package was used for data analysis.

  1. b) Also, the validity and reliability of the questionnaire is presented from line 160-175 as it follows:

3.4. The research instrument

The consistency of statistical data was analyzed, for the whole lot of 310 students, as well as for the data subsamples from the faculties (NUPSPA -CPR for 170 students, respectively BUES - IBE for 140 students).

Also, the validity and the reliability of the questionnaire was tested. The validity of the questionnaire that refers to the degree to which the proposed measurement measures what it wanted to measure [53] was ensured at the questionnaire conception phase through the checking of the draft questionnaire with a number of experts from higher education. Validity is also proved by demonstrating the veracity of the hypotheses [53,54]. The reliability of the questionnaire is verified by calculating the Alpha Cronbach for the two groups and for the entire population included in the study [55,56]. Calculated values for Alpha Cronbach reliability test are α = 0 .985 for whole data colectivity, α = 0.984 for NUPSPA -CPR colectivity and α = 0,983 for BUES – IBE colectivity. It can be concluded based on the Alpha Cronbach values, that there is a high level of statistical reliability, both for the entire data set and for the statistical data (two data subsets) analyzed at the university/faculty level.

Point 4: Finally, the conclusions need to be specific and linked to the results and the theoretical framework. Important also are the implications of the study for the educational actors involved, and also for the redefinition of the curricula and to give insights for policymakers.

Response 4: Section 6. Conclusions, limitations and future research was added to the present study as it follows (line 620, 623-649 and 656-667:

The theoretical framework for the study of global competences was developed based on the existing literature related to skills, abilities and competences and resulted in a typology of global competences that comprised six major global competences (international competences, personal competences, competences related to career management, workplace competences, theoretical competences and practical competences) each of them being built from a number of skills and abilities. All of them have an influence on employability according to the literature [10,29,33,36,37] and the present study also confirms that students consider all six categories of global competences as being necessary to activate in both national and international labour markets.

More specifically, it was observed an inclination of students studying in social studies to consider personal competences as being the most needed to build global competences, while students studying in economic field (specialization international business and economics) consider international competences as being the most important to build global competences. In the same vein, university/faculty activities oriented towards internationalization (both at home and abroad) were seen as being the main contributors to the development of skills needed in both national and international labour markets by students who study in the economic field. At the same time, extra-curricular activities and business and academia collaborations, among university’s/faculty’s activities, were seen by students who study in the social field, as main contributors to the development of global competences required in labour markets.

Also, the two groups of students presented statistically significant differences in their opinions about certain categories of global competences and included skills and abilities (International Competences, Theoretical Competences and Practical Competences), illustrating that they have rather different opinions about the contribution of different types of skills to the development of global competences in general, but also on the contribution of their respective university/faculty to the development of these categories of skills, abilities and competences. Other categories of competences (Personal Competences, Competences Related to the Career Management and Work-Place Competences) seem more similar in the eyes of the two groups of students.

The implications of the results of the present study are multifold: a) for higher education authorities and policy makers, based on the results of the present study it is recommended to support the development of a stronger international character of universities, faculties and study programs in order to develop global competences also required on international labour markets; b) for higher education institutions one main direct implication at organizational level would be the development of more collaborative study programs that would imply international partners both from the academia and the business environment, as appreciated by students and c) at student level one implication would be that they need to develop a combination of skills, abilities and competences in order to increase their employability in both national and international labour markets. The results of the study would imply the need redefining the curricula by including more international components, and this is a recommendation at higher education institution level, that it can also be encouraged through educational policy measures via designated programs.

Reviewer 2 Report

The article provides a well-developed research regarding the perception of students  of the neccesary competences for the national and international labour markets. The methodology and the objectives are clear from the beginning. The research questions are clear and well defined, the paper is accurate, and it contains a good literature review concerning the topic. The discussion of results is sound and well-conducted. The paper does read fluidly, and it is written in Standard English.

In order to improve this interesting article, I recommend the following improvements:

  • Introduction should be shorter: research justification, objectives and research questions, and description of the structure of the article.
  • Theoretical framework should be separate from the introduction.
  • I also recommend including a paragraph of limitations within the section 4 (Discussion and conclusions) and more extensive description of future research opportunities

In conclusion, I would like to thank the authors for a very interesting paper. Hope these comments and suggestions can help further their study.

Author Response

Response to Reviewer 2 Comments

Point 1: Introduction should be shorter: research justification, objectives and research questions, and description of the structure of the article. Theoretical framework should be separate from the introduction.

Response 1: The introduction was revised and separated from the Theoretical Framework. The introduction present now in addition to research justification, objectives and research question, the description of the structure of the article as it follows (line 43-67):

In a global world, economies and citizens must be updated with technology, global standards, global professions and global competences. Among the global professions, engineering represents the most standardized profession using mathematics, programming language and English language [1]. Consequently, the technical universities have some advantages in providing competences that are required by both national and international labour markets. In the field of socio – economic studies, the situation is different as there is less standardization involved. It is of interest to find out, from students’ point of view, what are the activities that universities specialized in social studies on the one hand and in economic studies, on the other hand, have to undertake to provide the global competences for their students, given less standardized rules and norms in applying the theory into practice in these two fields.

The Theoretical framework section will highlight the implications of the changing economic environment on global markets stakeholders’ activities and the role of internationalization for each stakeholder. The focus will be on the necessity for graduates, as future employees, to obtain global competences for both the national and the international labour markets. Also, the paper will briefly address the activities developed in universities (that are driven by the internationalization process) for providing global competences to their graduates.

The methodology section comprises three different subchapters in order to clarify the constructs of the present research: methods and statistical techniques used in research, the research variables and the hypotheses proposed to be tested. There are seven research questions that represent the starting point of the research and search for students’ perceptions on: skills and abilities needed for students to obtain global competences (RQ1); skills needed for them to activate in the national labour market (RQ2), or to activate in the international labour market (RQ3); activities specific of the universities which contribute to the development of the skills needed for the national labour market (RQ4), or contribute for skills needed for the international labour market (RQ5); skills and abilities perceived by students as being ensured by their university/faculty at a general level (RQ5) or about differences between opinions of students from the two different faculties with different specializations from the two universities (RQ7).

The Results section presents on one hand the mean calculation for each variable for the two groups of students and on the other hand validate the hypotheses. To validate the hypotheses the significant differences based on t-Test between students’ perceptions in the two universities/faculties are tested (Hyp. 1, Hyp. 2 and Hyp. 3) and the correlations between skills and university’s/faculty’s activities (Hyp. 4).

The Discussion and Conclusions section reveals aspects connected to global competences that relate to other studies’ findings and that show, in general, that the two groups of students with two study specializations have both similar, but also different opinions regarding the skills and abilities needed to gain global competences. They have a similar view related to personal competences, workplace competences and career management competences and different opinions related to international competences, theoretical competences and practical competences. In terms of competences provided by their universities/faculties for the national and international labour markets, the opinions converged only for the theoretical competences. All the other competences are perceived differently by students depending more on their specialization.

Point 2: I also recommend including a paragraph of limitations within the section 4 (Discussion and conclusions) and more extensive description of future research opportunities

Response 2: Section 6. Conclusions, limitations and future research was added to the present study as it follows (line 668-675 and 679-682):

As any research study, this study has also limitations. Among those, we can mention the inclusion of only two fields of studies (economic and social studies). Also, the fact that the study has been conducted in a specific national environment might affect results that also might be culturally related and not entirely generalizable at a larger level.

In the context of the existing limitations, we forward some suggestions for future research. For a more in-depth analysis, a confirmatory research could be conducted on the potential correlation between the ideal skills and abilities needed to build global competences, as perceived by students and the actual skills and abilities offered by different faculties and study programs. Also, in order to build a more comprehensive image about the development of global competences during the educational process in higher education institutions, of diverse profiles, the study can be extended through further research by including other fields of study (such as technical studies, medical studies and others) and identify a broader range of field related global competences. Another research direction can be the development of a cross-national sample, that would include also students from different countries in order to identify potential differences in various national context and also identify and re-inforce the common grounds at international level.

Thank you for your comments!

Reviewer 3 Report

This is a good paper and my few suggestions are to improve it even further.

I suggest that the literature and reference list needs a little updating and to be given an even more international flavour. To do this, the easiest thing would be to access some recent papers, not only for their content but also their extensive reference lists. In this regard, take a look at the work of O'Leary, S. on employability published in the Journal of Education and Work (2017) and also in the online latest articles of Studies in Higher Education (2019).

Overall, it is decent work that is worth publishing and this advice would I think help it further as the papers suggested address a range of variables in this field of research and will also direct you to many others.

Author Response

Response to Reviewer 3 Comments

Point 1: I suggest that the literature and reference list needs a little updating and to be given an even more international flavour. To do this, the easiest thing would be to access some recent papers, not only for their content but also their extensive reference lists. In this regard, take a look at the work of O'Leary, S. on employability published in the Journal of Education and Work (2017) and also in the online latest articles of Studies in Higher Education (2019).

Response 1:

The paper was enriched with new studies both in the Theoretical framework section and the Discussion section.

The new studies and articles that have been consulted and cited are marked with red color in the revised paper as it follows: 29, 32-37, 40-41, 46-56.

  1. Succi, C,; Canovi, M, Soft skills to enhance graduate employability: comparing students and employers’ perceptions. Studies in Higher Education 2019, DOI: 1080/03075079.2019.1585420, 1-14.
  2. Cleveland, S,; Cleveland, M, Leadership Competencies for Sustained Project Success. International Journal of Applied Management Theory and Research 2020, 2(1), 35-47.
  3. O’Leary, S, Graduates’ experiences of, and attitudes towards, the inclusion of employability-related support in undergraduate degree programmes; trends and variations by subject discipline and gender. Journal of Education and Work 2016, 30:1, 84-105.
  4. European Commission, Supporting Key Competences Development: Learning approaches and environments in school education. Conference Report, Brussels 12-13 November 2019. Available online: https://op.europa.eu/en/publication-detail/-/publication/37cc844f-4248-11ea-9099-01aa75ed71a1/language-en/format-PDF/source-search, (accessed 24 June 2020).
  5. Yari, N,; Lankut, E,; Alon, I,; Richter, N,F, Cultural Intelligence, global mindset, and cross-cultural competences: a systematic review using bibliometric methods. European J. International Mananagement 2020, 14(2), 210-250.
  6. Gedye, S,; Beaumont, E, The ability to get a job: student understandings and definitions of employability. Education + Training 2018, 60(5), 406-420.
  7. Monteiro, S,; Almeida, L,; Gomes, C,; Sinval, J, Employability profiles of higher education graduates: a person-oriented approach. Studies in Higher Education 2020, DOI: 1080/03075079.2020.1761785, Available online: http://repositorium.sdum.uminho.pt/handle/1822/65456 (accessed on 24 June 2020).
  8. Yorke, M,; Knight, P,T, Embedding employability into the curriculum. Learning and Employability Series I, The Higher Education Academy 2006. Available online: https://www.qualityresearchinternational.com/esecttools/esectpubs/yorkeknightembedding.pdf (accessed on 2 December 2018).
  9. Chhinzer, N,; Russo, A,M, An exploration of employer perceptions of graduate student employability. Education + Training 2018, 60(1), 104-120.
  10. Ala-Louko, R, Developing International Competence and Intercultural Communication Skills is an Investment in the Future. Lapland University of Applied Science, 2017, Available online: URN:NBN:fi:amk-2017092815485, (accessed on 23 June 2020).
  11. Singh, R,; Chawla, G,; Agarwal, S.; Desai, A, Employability and innovation: development of a scale. International Journal of Innovation Science 2017, 9(1), 20-37.
  12. Römgens, I,; Scoupe, R,; Beausaert, S, Unraveling the concept of employability, bringing together research on employability in higher education and the workplace. Studies in Higher Education 2019, DOI: 10.1080/03075079.2019.1623770, 1-16, Available online: https://www.tandfonline.com/doi/pdf/10.1080/03075079.2019.1623770, (accessed on 23 June 2020).
  13. Van der Heijden, B,I,J,M,; Notelaers, G,; Peters, P,; Stoffers, J,; De Lange, A,; Froehlich, D,E,; Van der Heijde, C,M, Development and validation of the short-form employability five-factor instrument. Journal of Vocational Behavior 2018, 106, 236-248.
  14. Caspersen, J,; Smeby, J-C, Placement training and learning outcomes in social work education. Studies in Higher Education 2020, DOI:10.1080/03075079.2020.1750583, 1-14, Available online: https://www.tandfonline.com/doi/pdf/10.1080/03075079.2020.1750583?needAccess=true (accessed on 25 June 2020)
  15. Gordon, L,E, Real research. Research Methods Sociology Students Can Use. SAGE Publications, Inc.: Thousand Oaks, USA, 2016, p.91.
  16. Creswell, J,W, Research Design. International Student Edition. Qualitative, Quantitative and Mixed Methods Approaches. SAGE Publications, Inc.: London, UK, 2014, p.163.
  17. Bolarinwa, O,A, Principles and methods of validity and reliability testing of questionnaires used in social and health science researches. Niger Postgrad Med J 2015, 22, 195-201
  18. Taber, K,S, The Use of Cronbach’s Alpha When Developing and Reporting Research Instruments in Science Education. Res Sci Educ 2018, 48, 1273–1296
  19. Deniz, M,S,; Alsaffar, A,A, Assessing the Validity and Reliability of a Questionnaire on Dietary Fibre-related Knowledge in a Turkish Student Population. Journal of Health Population and Nutrition 2013, 31(4), 497-503.

Thank you for your comments!
